# A Simple Clinical Scoring System to Determine the Risk of Pancreatic Cancer in the General Population

**DOI:** 10.3390/diagnostics14060651

**Published:** 2024-03-20

**Authors:** Dai Yoshimura, Mitsuharu Fukasawa, Yoshioki Yoda, Masahiko Ohtaka, Tadao Ooka, Shinichi Takano, Satoshi Kawakami, Yoshimitsu Fukasawa, Natsuhiko Kuratomi, Shota Harai, Naruki Shimamura, Hiroyuki Hasegawa, Naoto Imagawa, Yuichiro Suzuki, Takashi Yoshida, Shoji Kobayashi, Mitsuaki Sato, Tatsuya Yamaguchi, Shinya Maekawa, Nobuyuki Enomoto

**Affiliations:** 1Department of Gastroenterology, Faculty of Medicine, University of Yamanashi, Chuo 409-3898, Japan; dyoshimura@yamanashi.ac.jp (D.Y.); stakano@yamanashi.ac.jp (S.T.); skawakami@yamanashi.ac.jp (S.K.); mfukasawa@yamanashi.ac.jp (Y.F.); nkuratomi@yamanashi.ac.jp (N.K.); sharai@yamanashi.ac.jp (S.H.); nshimamura@yamanashi.ac.jp (N.S.); hirohasegawa@yamanashi.ac.jp (H.H.); nimagawa@yamanashi.ac.jp (N.I.); yuichirohs@yamanashi.ac.jp (Y.S.); tyoshida@yamanashi.ac.jp (T.Y.); shoji@yamanashi.ac.jp (S.K.); satom@yamanashi.ac.jp (M.S.); ytatsuya@yamanashi.ac.jp (T.Y.); maekawa@yamanashi.ac.jp (S.M.); enomoto@yamanashi.ac.jp (N.E.); 2Department of Gastroenterology, Japan Community Health Care Organization Yamanashi Hospital, Kofu 400-0025, Japan; 3Yamanashi Koseiren Health Care Center, Yamanashi 400-0035, Japan; y-yoda@y-koseiren.jp (Y.Y.); 8mohtaka20130423@gmail.com (M.O.); 4Department of Health Sciences, Faculty of Medicine, University of Yamanashi, Chuo 409-3898, Japan; tohoka@yamanashi.ac.jp; 5Department of Gastroenterology, Otsuki Municipal Central Hospital, Otsuki 401-0015, Japan

**Keywords:** pancreatic cancer, general population, scoring system, HbA1c, early-stage

## Abstract

This study aimed to develop and validate a simple scoring system to determine the high-risk group for pancreatic cancer (PC) in the asymptomatic general population. The scoring system was developed using data from PC cases and randomly selected non-PC cases undergoing annual medical checkups between 2008 and 2013. The performance of this score was validated for participants with medical checkups between 2014 and 2016. In the development set, 45 PC cases were diagnosed and 450 non-PC cases were identified. Multivariate analysis showed three changes in clinical data from 1 year before diagnosis as independent risk factors: ΔHbA1c ≥ 0.3%, ΔBMI ≤ −0.5, and ΔLDL ≤ −20 mg/dL. A simple scoring system, incorporating variables and abdominal ultrasound findings, was developed. In the validation set, 36 PC cases were diagnosed over a 3-year period from 32,877 participants. The AUROC curve of the scoring system was 0.925 (95%CI 0.877–0.973). The positive score of early-stage PC cases, including Stage 0 and I cases, was significantly higher than that of non-PC cases (80% vs. 6%, *p* = 0.001). The simple scoring system effectively narrows down high-risk PC cases in the general population and provides a reasonable approach for early detection of PC.

## 1. Introduction

Pancreatic cancer (PC) is a highly fatal disease with a 5-year survival rate of approximately 10% at the time of diagnosis [1]. According to the American Cancer Society, approximately 57,600 new PC cases were diagnosed in the USA in 2020 with an estimated 47,050 deaths, and it is expected to become the second leading cause of cancer death in approximately 10 years [1,2,3].

Recently, PC diagnosed at an early-stage has had a promising long-term prognosis. The Japan Study Group on the Early Detection of Pancreatic Cancer reported that the 10-year survival rate of cases with PC defined as Union for International Cancer Control (UICC) Stage 0 and I and smaller than 10 mm (TS1a) has reached 90% [4]. However, these cases account for only 0.8% of all PCs, making early diagnosis of PC very difficult [5]. Additionally, because 75% of PCs diagnosed at an early stage are asymptomatic [4], detecting early PC in the general population without symptoms is problematic.

PC is relatively uncommon and it would not be cost effective to screen for it in the general population [6]. A recommended conceptual framework for screening is a prospective two-sieve approach that includes the following three core phases: define, enrich, and find in a screening model [7,8]. The first step toward screening for asymptomatic, early-stage PC is to “define” high-risk groups for PC (first sieve), and the second step is to “enrich” these risk groups by using non-invasive imaging techniques (second sieve). Then, the use of more invasive technology to histologically confirm the diagnosis occurs within the third step, to “find” localized early-stage PC. Recently, characteristic imaging findings, including local parenchymal atrophy [9] and hypoechoic areas around the main pancreatic duct [10], as well as pathological diagnosis, such as serial pancreatic aspiration cytology (SPACE), have been reported as useful methods for early PC detection [11,12]. However, all of these imaging findings and methods can be used after the second sieve, and there is currently no useful method for the first sieve to identify high-risk groups for PC. Thus, this study aimed to develop a simple scoring system to determine high-risk groups for PC and to verify its usefulness in the general population undergoing medical checkups.

## 2. Methods

### 2.1. Study Population

This study was performed using data collected from participants who underwent annual medical checkups between April 2008 and March 2017 at the Yamanashi Koseiren Health Care Center. Participants were included in the analysis if they were aged at least 40 years old, underwent medical checkups for two consecutive years, and received examinations including height, weight, blood tests, and abdominal ultrasound. PC cases were identified with histopathological diagnosis by surgical resection or an endoscopic technique including endoscopic ultrasound-guided fine needle aspiration and biopsy (EUS-FNA, EUS-FNB) and pancreatic juice cytology with endoscopic retrograde cholangiopancreatography. Bile juice cytology was used for patients with obstructive jaundice. For PC cases, clinical information such as the location of tumor, UICC size, and treatment was collected. Control was defined as those who did not have PC for at least 2 years.

The study protocol was reviewed and approved by the Research Ethics Committee of the University of Yamanashi and Yamanashi Koseiren Health Care Center. The study was conducted in accordance with the declaration of Helsinki.

### 2.2. Data Collection

The clinical information obtained from the participants who had medical checkups included age, sex, height, weight, alcohol consumption, and smoking. Blood samples were collected in the morning after a 10 h overnight fast. Laboratory evaluation included red blood cell count, hemoglobin, hematocrit, platelet count, total protein, albumin, total bilirubin, alkaline phosphatase, aspartate aminotransferase (AST), and alanine aminotransferase (ALT), triglycerides, high-density lipoprotein cholesterol (HDL), low-density lipoprotein cholesterol (LDL), fasting blood glucose, and hemoglobin A1c (HbA1c). Body mass index (BMI) was calculated as the weight in kilograms divided by the square of the height in meters. Among those with a history of alcohol consumption, heavy drinkers were defined as those consuming >60 g of ethanol-equivalent alcohol per day. Smoking was categorized into current smokers, past smokers, and no smoking history. Among the findings from the abdominal ultrasounds, pancreatic tumor was defined as a hypo(iso)echoic or mixed hyper- and hypoechoic mass lesion. Main pancreatic duct dilatation was defined as a dilation of maximum diameter ≥3 mm at pancreatic body. Pancreatic duct diameter was measured from the beginning of the anterior wall echo to the beginning of the posterior wall echo, and recorded in millimeters after rounding off to the nearest integer. Pancreatic cyst was defined as a cystic lesion with maximum diameter ≥5 mm.

### 2.3. Scoring System Development

In this study, we conducted a two-step method to develop a simple scoring system that could identify the high-risk groups for PC. To establish the development set, we used the data from medical checkups between April 2008 and March 2013. We defined PC cases diagnosed during the period as the PC group, and randomly selected tenfold nonPC cases matched by age and sex as the control group, and we compared them to extract clinical data related to PC. We evaluated the following two values observed in the medical checkups: (A) clinical data at the time of PC diagnosis and (B) change in clinical data from 1 year before diagnosis. We utilized clinical data spanning two consecutive years, specifically using data from 8 to 16 months prior to diagnosis as the data from 1 year before PC diagnosis. We defined the change in values from 1 year before PC diagnosis as Δ values. For example, if the HbA1c level at the time of PC diagnosis was 6.8 and that from 1 year prior was 5.9, we expressed this as ΔHbA1c = 0.9. The cut-off values for clinical data at the time of diagnosis were used as the reference values. The cut-off values for the change in clinical data at 1 year before diagnosis were established by performing a receiver operating characteristic curve (ROC) analysis. Using the development sets for the PC and control groups, univariate and multivariate analyses were performed to extract clinical data related to PC, and the characteristics showing significant differences between the PC and control groups were included for further analysis. We developed a simple scoring system that is useful for the diagnosis of PC using the extracted clinical data. Scores were weighted based on the value of the beta coefficient of the logistic regression analysis. 

### 2.4. Validation of the Scoring System

To verify the external validity of the scoring system, 32,372 individuals who underwent medical checkups for at least two consecutive years between April 2014 and March 2017 were included in the validation set. We validated the scoring system using temporal validation and examined its diagnostic performance at each stage and compared it with that of CA19-9.

### 2.5. Statistical Analysis

Univariate analysis was performed using Fisher’s exact test for categorical variables and Mann–Whitney U test for continuous variables. Univariate and multivariate analyses were performed through logistic regression analysis to estimate the odds ratios and 95% confidence intervals (95% CI) to quantify the risk of clinical data associated with PC. When the variables used in the multivariate analysis contained missing values, a listwise deletion method was performed for a complete case analysis. The ROC analysis was performed to evaluate the diagnostic ability of the scoring system for PC diagnosis. The area under the curve (AUC) was calculated using 95%CI. Fisher’s exact test was performed when comparing the number of positive individuals on scores and CA19-9 in both PC cases and controls. A *p* value < 0.05 was chosen to indicate statistical significance.

All statistical analyses were performed using EZR (Version 1.54; Saitama Medical Center, Jichi Medical University, Saitama, Japan), which is a graphical user interface for R (R Foundation for Statistical Computing, Vienna, Austria) [13]. More precisely, it is a modified version of R Commander (Version 2.7-1) designed to add statistical functions frequently used in biostatistics. 

## 3. Results

From April 2008 to March 2017, 222,305 individuals (918,206 person years) underwent medical checkups. Of these, participants younger than 40 years, cases with lacking physical and blood findings, cases without abdominal ultrasonography, and cases without checkups for two consecutive years were excluded. Altogether, 91,045 participants with medical checkups were finally included in the study. 

To establish the development set, we used data from those who underwent medical checkup between April 2008 and March 2013. Among the 58,179 participants who underwent medical checkups during this period, 45 cases were diagnosed with PC. From the nonPC cases during the same period, 450 age- and sex-matched nonPC cases were randomly selected as controls (nonPC group). For the validation set, 32,372 participants who had undergone medical checkups between April 2014 and March 2017 were included (Figure 1).

### 3.1. Clinical Characteristics of Individuals in the Test Set

Table 1 shows the clinical characteristics of the PC and nonPC groups in the development set. The median age was 70 years, 49% cases were male, median BMI was 22.6 kg/m^2^, and there was no significant difference in the proportion of heavy drinkers and current smokers between the two groups. 

In the PC group, 19 cases (42.2%) were located in the pancreatic head, and the tumor size was ≤20 mm in 11 cases (24.4%). Four cases (8.9%) were Stage 0 or I, twenty-two cases (48.9%) were Stage II, and fifteen cases (33.3%) were diagnosed as Stage IV disease. As for treatment, surgery and chemotherapy were performed in 25 cases (55.6%) and 17cases (37.8%), respectively (Table 2).

### 3.2. Clinical Data Associated with PC and the Development of a Simple Clinical Scoring System

Univariate and multivariate analyses were performed to extract the following factors associated with PC: (A) clinical data at the time of PC diagnosis and (B) change in clinical data from 1 year before diagnosis. The multivariate analyses, except that of glucose, which was confounded by HbA1c, showed that ΔBMI [odds ratio (OR) 2.42, 95% confidence interval (CI) 1.21–4.85, *p* = 0.013], ΔLDL (OR 2.31, 95%CI 1.01–5.31, *p* = 0.048), and ΔHbA1c (OR 8.29, 95% CI 3.39–20.30, *p* < 0.001) were independent risk factors associated with PC (Table 3).

Based on the logistic regression model, a simple scoring system to determine the high-risk group for PC was established with three clinical factors. ΔBMI, ΔLDL, and ΔHbA1c were classified into three categories for each value, and the scores were defined. The β value of ΔBMI was the smallest, and the β values of the other variables were divided by the minimum regression coefficient and multiplied by two to obtain the rounded results as the scores of each variable (regression coefficients for ΔBMI, ΔLDL, and ΔHbA1c were 0.88, 0.84, and 2.11, respectively). To efficiently narrow down the high-risk group for PC, the abdominal ultrasonographic findings were employed and incorporated into the score. A score was assigned to the presence or absence of pancreatic mass, pancreatic duct dilation, and pancreatic cyst. Finally, a simple scoring system was created, which was named the ΔPC screening score (Table 4). According to the ROC curve, the AUC was 0.900 (95% CI 0.840–0.961), and the sensitivity and specificity values were 64.4% and 95.3%, respectively, when six points were used as the cut-off (Figure 2a).

### 3.3. Validation of the ΔPC Screening Score

The usefulness of the ΔPC screening score for extracting the high-risk group for PC was verified using the validation set of 32,372 participants who had undergone medical checkups for two consecutive years between April 2014 and March 2017. Of these participants, 36 individuals were diagnosed with PC. The ROC curve for the ΔPC screening score in the validation set showed that the AUC was 0.925 (95% CI 0.877–0.973), and the sensitivity and specificity values were 77.8% and 93.9%, respectively (Figure 2b). 

Figure 3 shows the diagnostic performance of the ΔPC screening score and CA19-9 according to the UICC stage of PC. The positive rate of CA19-9 in Stages 0 or I, II, III or IV was 20.0%, 53.3%, and 68.8%, respectively, and increased with cancer progression. The positive rate was significantly higher in patients with Stage ≥II diseases than in the controls, but the rate was not significantly different between patients with Stage ≥II diseases and those with Stage 0 or I disease. Contrarily, the positive rate of ΔPC screening score in Stages 0 or I, II, III or IV was 80.0%, 73.3%, and 81.3%, respectively, which was significantly higher than that of the control group in all stages, including Stage 0 or I.

## 4. Discussion

We developed and validated a simple scoring system to extract the high-risk group for PC from the general population. Using data from participants who had undergone health checkups for two consecutive years, a multivariate analysis of the factors associated with PC extracted change values (Δ) for BMI, LDL, and HbA1c from 1 year before diagnosis. The scoring system showed a high diagnostic performance with AUROC of 0.900 and 0.925 for the development and validation sets, respectively. Furthermore, compared with CA19-9, the ΔPC screening score was useful in identifying early PC.

Screening for PC in the general population is not recommended due to its low incidence and poor cost-effectiveness. The International Cancer of the Pancreas Screening consortium of the World Health Organization recommends screening for PC only in high-risk carriers with a lifetime risk of at least 5% or a five-fold relative risk [6]. However, such high-risk individuals are limited to a specific population with hereditary predisposition or pancreatic cystic lesions. For early diagnosis of PC, establishing a screening method that determines the high-risk groups from the general population is necessary. Recently, various biomarkers have been reported to be useful for the diagnosis of PC [14,15,16,17,18,19,20]. However, their application in medical checkups or clinical practice is limited. To detect PCs in the general population, it is important to establish a screening method that is simple, non-invasive, can be performed during medical checkups, and can detect early stage lesions. In this study, we focused on the temporal changes in clinical data of patients who had undergone medical checkups and created a scoring system that had a high diagnostic performance. This indicates that it may be possible to narrow down the high-risk group for PC by encouraging patients to have medical checkups for two consecutive years, even if they do not have risk factors, such as hereditary predisposition or pancreatic cysts. We also evaluated the external validity of the scoring system using cases from different time periods from the development set to assess its usefulness. External validity is generally considered to have a stronger significance than internal validity in validating predictive models [21]. 

The three factors used in the present scoring system have been reported to be associated with PC in the literature. A meta-analysis reported that diabetes mellitus was associated with an increased risk of PC; in particular, there was a 5.38 times higher relative risk for patients who had diabetes for <1 year [22]. Based on such observations, new-onset diabetes (NOD) has been suggested as a possible target for PC screening. Sharma et al. proposed the ENDPAC score, which is based on age at NOD and change in blood glucose levels and weight 1 year before diagnosis [23]. For an ENDPAC score of ≥3, the incidence of PC within 3 years was as high as 3.6%, with a sensitivity of 78% and a specificity of 80%. The focus on changes in glucose tolerance is interesting; however, its application is limited to patients with NOD. In this study, a ΔHbA1c of at least 0.3 was extracted as a factor associated with PC. It is important to focus on the changes in HbA1c not only in patients with NOD but also in participants without glucose intolerance since some cases show changes in the normal range. Regarding the association between lipid abnormalities and PC, hyperlipidemia and hypercholesterolemia are not considered direct factors for PC [24]. However, Raghuwansh et al. reported a decrease in serum lipids, body weight, abdominal subcutaneous fat (SAT) with preserved visceral adipose tissue, and muscle 18 months before PC diagnosis and found that the overexpression of uncoupling protein1 (UCP1) in SAT exposed the patients to PC exosomes [25]. It is suggested that focusing on the changes in lipids may contribute to the early diagnosis of PC. Additionally, the abdominal ultrasound findings in the medical checkups were incorporated into the score. Abdominal ultrasound is a simple and non-invasive examination that is considered useful in screening and medical examinations for PC, with a sensitivity of 88% and specificity of 94% reported from a meta-analysis [26]. Although there is the disadvantage that the imaging performance is likely to vary depending on the patient’s body shape and the examiner’s skill, an evaluation together with changes in BMI, LDL, and HbA1c may be useful as part of the initial screening to extract the high-risk group for PC from the general population.

Recently, it has been reported that the prognosis of early-stage PC is favorable. Therefore, diagnosis at an earlier stage is the key to prolonging the prognosis of PC. The ΔPC screening score developed in this study showed a high positive rate of 80% even in Stage 0–I cases, and was considered useful as a means of narrowing down early stage PC in the general population. Additionally, 58% of the PC cases in the development set were diagnosed with Stages 0–II in resectable conditions. It has been reported that 10–15% of PC cases are localized and resectable at the time of diagnosis [1], and the resection rate in this study was relatively high. This may be due to the fact that most of the cases were detected in an asymptomatic state since the present study included individuals who had undergone medical checkups for two consecutive years. Moreover, it is important to identify the high-risk groups for PC from the asymptomatic general population who have undergone medical checkups.

The present study has several limitations. First, this was a single-center retrospective study. Second, there was confounding regarding patient background because the data from the health care center lacked information on comorbidities, such as dyslipidemia, diabetes, malignant diseases of other organs, and medication history. Third, individuals who had not undergone medical checkups for two consecutive years were not included, which may have introduced a selection bias. Therefore, prospective validation in other cohorts is desirable to evaluate the usefulness of our scoring system in actual clinical practice.

## 5. Conclusions

In conclusion, the ΔPC screening score, a simple scoring system based on changes in BMI, LDL, HbA1c, and abdominal ultrasound findings, is a useful tool for determining the group at risk of PC from the general population. By undergoing checkups for two consecutive years, it may be possible to diagnose early stage PC with the potential for long-term survival.

## Figures and Tables

**Figure 1 diagnostics-14-00651-f001:**
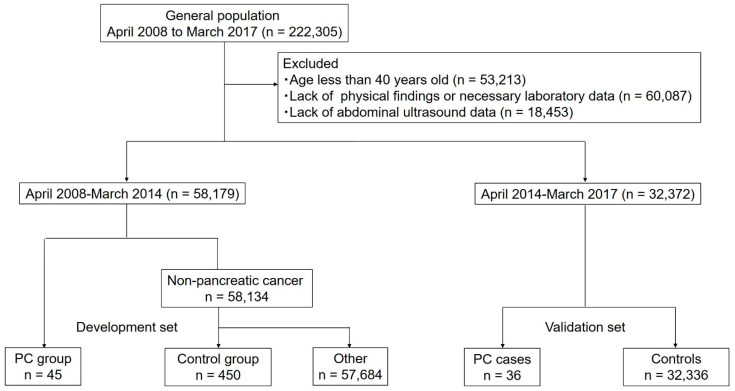
Flowchart of this study.

**Figure 2 diagnostics-14-00651-f002:**
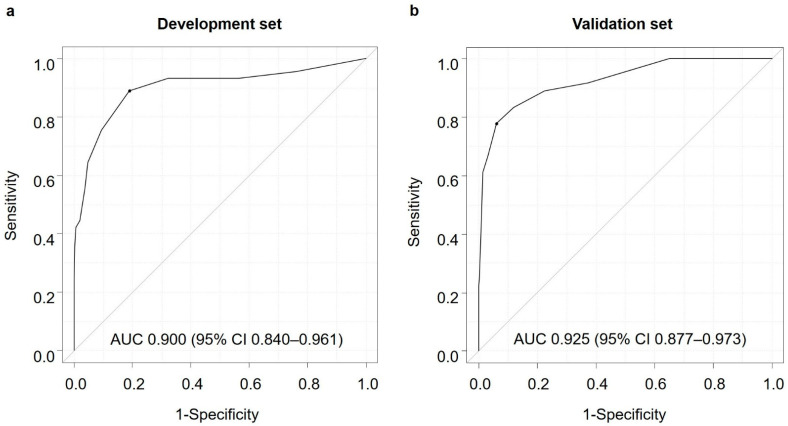
Receiver operating characteristic (ROC) curves of the ΔPC screening score for test and validation sets. (**a**): ROC curve for test set (**b**): ROC curve for validation set.

**Figure 3 diagnostics-14-00651-f003:**
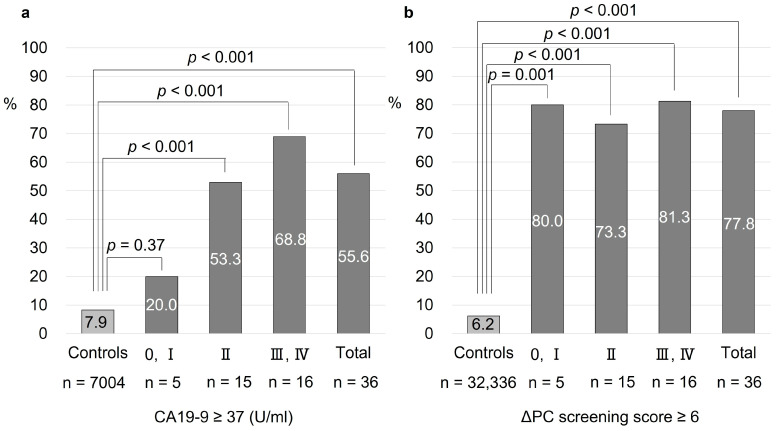
The proportion of PC cases exhibiting positive of ΔPC screening score and CA19-9 at each stage. (**a**): The proportion of PC cases that CA19-9 is more than 37 (U/mL) at each stage. (**b**): The proportion of PC cases that ΔPC screening score is more than 6 points at each stage.

**Table 1 diagnostics-14-00651-t001:** Clinical characteristics of individuals in the development set.

	PC Group		Control Group		
	n = 45		n = 450		*p* Value
Age (years)	70	(64–77)	70	(64–77)	1
Sex (male)	22	(48.9)	220	(48.9)	1
BMI	22.6	(20.7–26.0)	22.6	(20.6–24.5)	0.75
Alcohol consumption (g/day)					
≥0, <20	36	(80.0)	400	(88.9)	0.091
≥20, <60	8	(17.8)	47	(10.4)	0.14
≥60	1	(2.2)	3	(0.7)	0.32
Smoking					
Never	31	(68.9)	278	(61.8)	0.42
Past	12	(26.7)	136	(30.2)	0.73
Current	2	(4.4)	36	(8.0)	0.56

Results are presented as numbers with percentages in parenthesis for qualitative data or as medians with IQR in parenthesis for quantitative data. PC, pancreatic cancer; BMI, body mass index; IQR, interquartile range.

**Table 2 diagnostics-14-00651-t002:** Clinical characteristics of pancreatic cancer patients in the development set.

	Development Set	
	n = 45	
Location		
Ph	19	(42.2)
Pb	17	(37.8)
Pt	9	(20.0)
Tumor size (mm)		
≥0, ≤20	11	(24.4)
>20, ≤40	28	(62.2)
>40, ≤60	3	(6.7)
>60	3	(6.7)
UICC Stage		
0, Ⅰ	4	(8.9)
Ⅱ	22	(48.9)
Ⅲ	4	(8.9)
Ⅳ	15	(33.3)
Treatment		
Surgery	25	(55.6)
Chemotherapy	17	(37.8)
Best supportive care	3	(6.7)

UICC, Union for International Cancer Control.

**Table 3 diagnostics-14-00651-t003:** Univariate and multivariate analyses of clinical data associated with pancreatic cancer.

		PC Group		Control Group		Univariate	Multivariate		
		n = 45		n = 450		*p* Value	*p* Value	OR	(95%CI)
A. Clinical data at diagnosis of PC									
BMI	<18.5	4	(8.9)	41	(9.1)	0.77			
BMI	≥25	12	(26.7)	89	(19.8)	0.33			
RBC	≤4.0 (×10^3^/μL)	10	(22.2)	63	(14.0)	0.18			
Hemoglobin	≤12.0 (g/dL)	5	(11.1)	43	(9.6)	0.79			
Hematocrit	≤36.0 (%)	4	(8.9)	41	(9.1)	1			
Platelet count	≤140 (×10^3^/μL)	1	(2.2)	44	(9.8)	0.35			
Albumin	≤3.8 (mg/dL)	2	(4.4)	43	(9.6)	0.57			
Trigliceride	≤150 (mg/dL)	6	(13.3)	51	(11.3)	0.63			
HDL	≤60 (mg/dL)	3	(6.7)	30	(6.7)	1			
LDL	≥110 (mg/dL)	9	(20.0)	103	(22.9)	0.85			
AST	≥35 (U/L)	6	(13.3)	32	(7.1)	0.14			
ALT	≥35 (U/L)	4	(8.9)	30	(6.7)	0.53			
ALP	≥350 (U/L)	2	(4.4)	24	(5.3)	1			
Glucose	≥110 (mg/dL)	23	(51.1)	97	(21.6)	<0.001			
HbA1c	≥6.5 (%)	13	(28.9)	39	(8.7)	<0.001	0.78	1.15	(0.43–3.07)
B. Change in clinical data from one year before diagnosis (Δ)									
ΔBMI	≤−0.5	20	(44.4)	114	(25.3)	0.008	0.013	2.42	(1.21–4.85)
ΔRBC	≤−0.2 (×10^3^/μL)	17	(37.8)	91	(20.2)	0.013	0.065	2.04	(0.96–4.36)
ΔHemoglobin	≤−0.5 (g/dL)	17	(37.8)	110	(24.4)	0.072			
ΔHematocrit	≤−1.0 (%)	20	(44.4)	141	(31.3)	0.094			
ΔPlatelet count	≤−20 (×10^3^/μL)	9	(20.0)	68	(15.1)	0.39			
ΔAlbumin	≤−0.3 (mg/dL)	6	(13.3)	63	(14.0)	1			
ΔTrigliceride	≤−10 (mg/dL)	23	(51.1)	169	(37.6)	0.079			
ΔHDL	≤−10 (mg/dL)	1	(2.2)	27	(6.0)	0.50			
ΔLDL	≤−20 (mg/dL)	14	(31.1)	48	(10.7)	<0.001	0.048	2.31	(1.01–5.31)
ΔAST	≥5 (U/L)	7	(15.6)	71	(15.8)	1			
ΔALT	≥5 (U/L)	12	(26.7)	65	(14.4)	0.049	0.16	1.82	(0.79–4.17)
ΔALP	≥10 (U/L)	12	(26.7)	159	(35.3)	0.32			
ΔGlucose	≥10 (mg/dL)	12	(26.7)	37	(8.2)	<0.001			
ΔHbA1c	≥0.3 (%)	18	(40.0)	30	(6.7)	<0.001	<0.001	8.29	(3.39–20.30)

PC, pancreatic cancer; OR, odds ratio; 95%CI, 95% confidence interval; BMI, body mass index; RBC, red blood cells; ALP, alkaline phosphatase; HDL, high density lipoprotein cholesterol; LDL, low density lipoprotein cholesterol; AST, aspartate aminotransferase; ALT, alanine aminotransferase; ALP, alkaline phosphatase; HbA1c, Hemoglobin A1c.

**Table 4 diagnostics-14-00651-t004:** Parameters for ΔPC screening score.

	Score	Score Range
ΔLDL (mg/dL)		0–2
≤−20	2	
>-20, ≤−10	1	
>−10	0	
ΔHbA1c (%)		0–5
≥0.3	5	
≥0.1, ˂0.3	2	
˂0.1	0	
ΔBMI		0–2
≤−0.5	2	
>−0.5, ≤0	1	
>0	0	
Abdominal ultrasound		0–14
Pancreatic tumor	6	
Main pancreatic duct dilation	6	
Pancreatic cyst	2	
Total		0–23

LDL, low density lipoprotein cholesterol; BMI, body mass; index; HbA1c, Hemoglobin A1c.

## Data Availability

The authors declare that the data for this research are available from the correspondence authors upon reasonable request.

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
