# Peer review of "A Simple Clinical Scoring System to Determine the Risk of Pancreatic Cancer in the General Population"

_diagnostics, 2024, doi:10.3390/diagnostics14060651_

Round 1

Reviewer 1 Report

Comments and Suggestions for Authors

The manuscript entitled “A simple clinical scoring system to determine the risk of pan2 creatic cancer in the general population” is a research paper determining a scoring system based on the difference of BMI, LDL, HbA1c and ultrasound findings in two consecutive years to detect early stage patients with pancreatic ductal adenocarcinoma. The authors employed a development and a validation set and the results are very encouraging. Importantly, this proposed is non-invasive approach with low cost and high benefit. Besides, to develop a scoring system that is reproducible and straightforward for early detection of pancreatic cancer is crucial and therefore, this manuscript is suitable for publication in Diagnosis. Below are a few comments that could further improve manuscript quality:

1)        Table 1: The Age, Sex and BMI: could the authors check the values written without parenthesis

2)        The authors need to define the markers ΔBMI, ΔLDL and ΔHbA1c in the Material and Methods. What is the appropriate time frame (i.e. two consecutive years)? In the development and validation set how did the authors assess these parameters? Could they provide an example. This approach to be reproducible needs to be precisely described.

Author Response

Thank you very much for taking the time to review this manuscript. Please find the detailed responses in track changes in the re-submitted files.

Reviewer 2 Report

Comments and Suggestions for Authors

Interesting article looking at a scoring method for early pancreatic detection

the authors note that the low incidence does not support the cost effectiveness of a formal screening program. They then disucuss changes in biochemical and othe parameters as a scoring method which can be used as a surrogate screen in those patients who are undergoing these tests as part of a general health check up and that looks to be valid.

I am then concerned that suddenly a more invasive investigation such as an Ultrasound is introduced as a screen when this would not appear to be part of the general checkup. This brings in a significant resource allocation and manpower allocation which suddenly becomes part of the general checkup and then becomes more a of a formal screening program. If ultrasound is a routine part of a general chack up, this is certainly not the experience of most western/australasian/americas routine checkups and the applicability of the scoring system is thus limited outside of this centre 

It would be more useful if the authors reviewed their data to see if a more directed method of ultrasound screening of patients with abnormal investigations andf then who moved onto ultrasound investigations. This would give a far more realistic clinical experience which would have international clinical applicability

Othewise the language in the language was clumsy with a confusing choice of words eg last word on line 53 would be much clearer if the word problematic was used rather than essential and i would suggest review by a native english speaker to adjust some of the idiom

Comments on the Quality of English Language

as noted above, requires review by native english speaker for review of idiom

Author Response

(The authors gave the same response as above.)

Round 2

Reviewer 2 Report

Comments and Suggestions for Authors

authors have met my comments to my satisfaction

Author Response

Thank you for taking the time to read our response letter. We are pleased that our comments met your expectations. We hope that our paper will be appreciated by many more people.